# Interactions between Human Gut Microbiome Dynamics and Sub-Optimal Health Symptoms during Seafaring Expeditions

Zheng Sun,[a,c] Meng Zhang,[a,b] Min Li,[a,b] Yogendra Bhaskar,[c] Jinshan Zhao,[d] Youran Ji,[e] Hongbing Cui,[f] Heping Zhang,[a,b] Zhihong Sun[a,b]

[a]Key Laboratory of Dairy Biotechnology and Engineering, Ministry of Education, Inner Mongolia Agricultural University, Huhhot, China
[b]Key Laboratory of Dairy Products Processing, Ministry of Agriculture and Rural Affairs, Inner Mongolia Agricultural University, Huhhot, China
[c]Single-Cell Center and Shandong Key Laboratory of Energy Genetics, Qingdao Institute of BioEnergy and Bioprocess Technology, Chinese Academy of Sciences, Qingdao, Shandong, China
[d]College of Animal Science, Qingdao University, Qingdao, Shandong, China
[e]Medical Department, 971 Hospital, Qingdao, Shandong, China
[f]Department of Emergency, Qilu Hospital (Qingdao), Cheeloo College of Medicine, Shandong University, Qingdao, Shandong, China

Zheng Sun, Meng Zhang and Min Li contributed equally to this article. Author order was determined by drawing straws.

**ABSTRACT** During long ocean voyages, crew members are subject to complex pressures from their living and working environment, which lead to chronic diseases-like sub-optimal health status. Although the association between dysbiotic gut microbiome and chronic diseases has been broadly reported, the correlation between the sub-optimal health status and gut microbiome remains elusive. Here, the health status of 77 crew members (20–35 years old Chinese, male) during a 135-day sea expedition was evaluated using the shotgun metagenomics of stool samples and health questionnaires taken before and after the voyage. We found five core symptoms (e.g., abnormal defecation frequency, insomnia, poor sleep quality, nausea, and overeating) in 55 out of 77 crew members suffering from sub-optimal health status, and this was termed "seafaring syndrome" (SS) in this study. Significant correlation was found between the gut microbiome and SS rather than any single symptom. For example, SS was proven to be associated with individual perturbation in the gut microbiome, and the microbial dynamics between SS and non-SS samples were different during the voyage. Moreover, the microbial signature for SS was identified using the variation of 19 bacterial species and 26 gene families. Furthermore, using a Random Forest model, SS was predicted with high accuracy (84.4%, area under the concentration-time curve = 0.91) based on 28 biomarkers from pre-voyage samples, and the prediction model was further validated by another 30-day voyage cohort (accuracy = 83.3%). The findings in this study provide insights to help us discover potential predictors or even therapeutic targets for dysbiosis-related diseases.

**IMPORTANCE** Systemic and chronic diseases are important health problems today and have been proven to be strongly associated with dysbiotic gut microbiome. Studying the association between the gut microbiome and sub-optimal health status of humans in extreme environments (such as ocean voyages) will give us a better understanding of the interactions between observable health signs and a stable versus dysbiotic gut microbiome states. In this paper, we illustrated that ocean voyages could trigger different symptoms for different crew member cohorts due to individual differences; however, the co-occurrence of high prevalence symptoms indicated widespread perturbation of the gut microbiome. By investigating the microbial signature and gut microbiome dynamics, we demonstrated that such sub-optimal health status can be predicted even before the voyage. We termed this phenomenon as "seafaring syndrome." This study not only provides the potential strategy for health management in extreme environments but also can assist the prediction of other dysbiosis-related diseases.

Address correspondence to Zhihong Sun, sunzhihong78@163.com.

The authors declare no conflict of interest.

**KEYWORDS** seafaring syndrome, ocean going voyage, sub-optimal health, gut microbiome, prediction model

The challenging conditions encountered during ocean voyages increase the risk of poor health in crew members who have to live in extreme conditions such as cramped living quarters with high humidity, high salinity, intense UV radiation, insufficient fruit and vegetables, and lack of physical exercise (1). Harsh living and working environments can cause crew members a number of physical and psychological symptoms, which are indicative of sub-optimal health (2–7) and closely associated with several diseases, such as cardiovascular diseases (8, 9), scurvy (10), oral diseases (11, 12), digestive system diseases, musculoskeletal disorders and circulatory diseases (13). Moreover, key factors such as isolation from family, limited activity space and an unhealthy diet are the main causes of mental stress (7, 14). Scientists and staff onboard during expeditions may experience psychological stress and subjective fatigue which are significantly related to the environment at sea (2, 15).

The gut microbiome interacts with the human immune system and plays a key role in human physical and mental health (16, 17). Previous studies have also shown that long-term living in a closed or semi-closed environment (e.g., a space capsule or small bunk beds) profoundly affects the gut microbiome (18–20). Although much attention has been paid to manage the health of crew members during long sea voyages, research on the roles of and interactions with the human microbiome remains limited. Two studies found a significant changes in oral microbial diversity after voyages (21, 22), while another study reported that probiotics can maintain homeostasis of the gut microbiome of crew members during a 1-month sea voyage (23). However, the following questions remain unanswered: (i) how the gut microbiome responds to ocean voyages (i.e., voyages of longer than 5 months); (ii) what are the gut microbiome dynamics in crew members with sub-optimal symptoms during ocean expeditions?

In this study, association between the gut microbiome and the health status of 77 crew members during a 135-day sea expedition was evaluated using whole metagenome sequencing (WMS) of stool samples and pre- and post-voyage health questionnaires. We found five core symptoms were experienced by crew members who were suffering with sub-optimal health, and we described this phenomenon as Seafaring Syndrome (SS). Next, the crew members' gut microbiome was influenced by SS expressed by multiple symptoms, which was proved to be a key factor for individual perturbation in the gut microbiome during the voyage. Finally, we validated a high accuracy model based on the microbial signature of SS to predict the likelihood of SS even before sea voyages.

## RESULTS

**Investigation of physical and psychological symptoms after a 135-day sea expedition.** The major symptoms during the ocean voyage were investigated using the questionnaires at the beginning (day 1) and end (day 135) of the voyage. To avoid potential confounding factors of age, gender and nationality, only data from the predominant group of crew members were analyzed (male, Han nationality, China; age 20 to 35). The questionnaire consisted of three parts (Method): (i) physical indicators, (ii) psychological indicators and more specifically, and (iii) defecation related indicators.

In terms of physical symptoms (Fig. 1a), the highest incidence among crew members was for backache (46.8%) and headache (44.2%) which increased by 40.3% and 39.0%, respectively, compared to the baseline values (6.5 and 5.2%, respectively). The incidence of muscular soreness (35.1%) and stomachache (33.8%) were the next most common symptoms, with an increase of 27.3 and 29.9% respectively compared to the baseline (7.8 and 3.9%, respectively). However, the body mass index (BMI) of all crew members showed no significant difference during the voyage.

Major psychological problems were reported by crew members (Fig. 1b). Poor sleep quality (77.9%) and insomnia (66.2%) were most evident among crew members, while few had sleeping problems before the voyage. More than half of the crew members

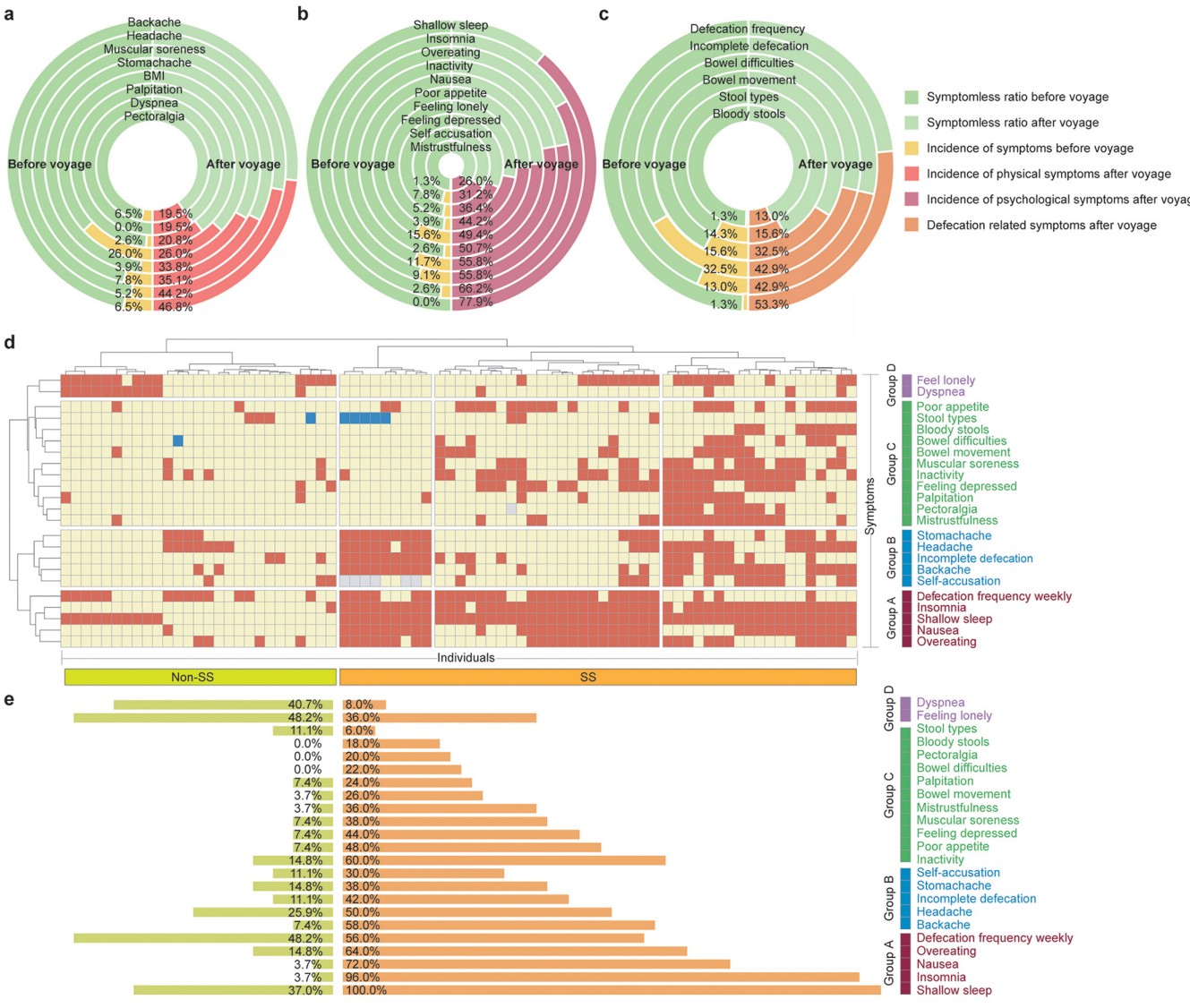

**FIG 1** Sub-optimal health symptoms during ocean voyage and characteristics of seafaring syndrome. Changes in self-assessed indicators of physical (a), psychological (b) and defecation-related (c) symptoms. The left half of each circle refers to the incidence of symptoms before travel while the right half of each circle refers to the incidence of symptoms after the 135-day voyage. (d) Heatmap showing the change of different health indicators (symptoms) adjusted by clustering individual results. Red squares represent symptoms that increased in incidence after the voyage, yellow squares represent symptoms that showed for no change and blue squares represent symptoms that decreased in incidence after the voyage). Four groups of symptoms were classified, and Group A was the core group of symptoms representative of SS. (e) Crew members clustered in orange represent individuals suffering with SS while the light green indicates 'symptomless' individuals without SS. Comparison of the incidence of symptoms between SS and non-SS groups are clustered by groups (group A through group D).

experienced overeating (55.8%), inactivity (55.8%) and nausea (50.7%) which showed increases of 46.8%, 44.2% and 48.1%, respectively compared before voyage results. These symptoms suggest a high level of mental stress due to the ocean voyage.

More specifically, we found that changes in defecation was a common symptom for most crew members (Fig. 1c). More than half (53.3%) of the crew members reported an abnormal frequency of defecation (weekly) compared to only 1.3% at baseline (defecation (weekly)). In addition, 42.9% of crew reported incomplete defecation (a 29.9% increase from baseline) and/or constipation (32.5%) that did not result in defecation (a 16.9% increase from baseline).

**Definition of seafaring syndrome.** Although the extreme living and working environments endured by the crew were very similar, different individuals experienced different symptoms. To explore the correlation among the symptoms, we conducted

**TABLE 1** Permanova test on the gut microbiome using different physical and psychological health indicators (symptoms) as factors

| Factors | BCD | | rJSD | |
|---|---|---|---|---|
| | Adonis.F | Adonis.P | Adonis.F | Adonis.P |
| Host ID | 2.689 | 0.001 | 2.523 | 0.001 |
| Seafaring syndrome | 1.946 | 0.020 | 1.810 | 0.024 |
| Time points | 1.762 | 0.043 | 1.481 | 0.063 |
| Self-accusation | 1.588 | 0.072 | 1.375 | 0.100 |
| Feeling depressed | 1.585 | 0.054 | 1.424 | 0.083 |
| Dyspnea | 1.519 | 0.086 | 1.418 | 0.085 |
| Bloody stool | 1.464 | 0.081 | 1.354 | 0.096 |
| Bowel difficulties | 1.405 | 0.136 | 1.394 | 0.104 |
| Backache | 1.324 | 0.147 | 1.186 | 0.189 |
| Defecation frequency | 1.290 | 0.159 | 1.137 | 0.245 |
| Insomnia | 1.181 | 0.222 | 1.082 | 0.314 |
| Palpitations | 1.169 | 0.257 | 1.142 | 0.229 |
| Inactivity | 1.167 | 0.259 | 1.199 | 0.191 |
| Poor sleep quality | 1.157 | 0.237 | 1.042 | 0.347 |
| BMI | 1.130 | 0.282 | 1.072 | 0.322 |
| Nausea | 1.115 | 0.298 | 0.967 | 0.487 |
| Overeating | 0.978 | 0.431 | 1.006 | 0.379 |
| Stool type | 0.934 | 0.497 | 0.863 | 0.649 |
| Mistrustfulness | 0.913 | 0.527 | 0.950 | 0.488 |
| Pectoralgia | 0.913 | 0.533 | 0.904 | 0.572 |
| Headache | 0.903 | 0.552 | 1.008 | 0.383 |
| Poor appetite | 0.858 | 0.603 | 0.974 | 0.442 |
| Incomplete defecation | 0.855 | 0.594 | 0.867 | 0.616 |
| Constipation | 0.845 | 0.631 | 0.961 | 0.456 |
| Feeling lonely | 0.841 | 0.637 | 0.841 | 0.679 |
| Stomach ache | 0.811 | 0.662 | 0.917 | 0.540 |
| Muscular soreness | 0.565 | 0.955 | 0.701 | 0.912 |

cluster analysis on the change of physical, psychological and defecation indicators of crew members during the ocean voyage (Fig. 1d and e, Method). Based on the these symptoms, crew members were clustered into four distinct groups: (i) Group A: the crew members, which were suffering from abnormal defecation frequency (weekly), insomnia, poor sleep quality, nausea and overeating; (ii) Group B: in addition to the symptoms of Group A, a few crew members also suffered from stomachache, headache, incomplete defecation, backache and self-accusation; (iii) Group C: in addition to the symptoms of Group A, a part of crew members also suffered from number of abnormalities, such as poor appetite, abnormal stool type (e.g., watery stool), bloody stool, bowel difficulties, constipation (without defecation), muscular soreness, feeling of tiredness leading to inactivity, feeling depressed, palpitations, pectoralgia and mistrustfulness; (iv) Group D: the crew members that showed only symptoms of loneliness and dyspnea (i.e., excluding group A through group C symptoms). We considered the co-appearance of symptoms of Group A (e.g., abnormal defecation frequency, insomnia, poor sleep quality, nausea and overeating) as the core characteristics of this newly discovered sub-optimal health status for long voyage crew members, which we named "seafaring syndrome" (SS). In addition to the core symptoms (that almost every SS crew suffered from), SS crew members were also prone to suffer from symptoms of Group B and Group C (Fig. 1e).

**Association between the gut microbiome and SS.** To explore any relationship between SS and the gut microbiome, we collected stool samples and conducted WMS at two time points, i.e., at day 1 and day 135 (Method). Firstly, to determine the effect size of SS on the gut microbiome of crew members, Permanova test was performed. We found the influence of SS as a factor on the gut microbiome was significant (Table 1B–D: $F = 1.946$, $P = 0.020$; root Janson Shannon distance [rJSD]: $F = 1.810$, $P = 0.024$), and its significance was greater than the influence of time point (Bray-Curtis

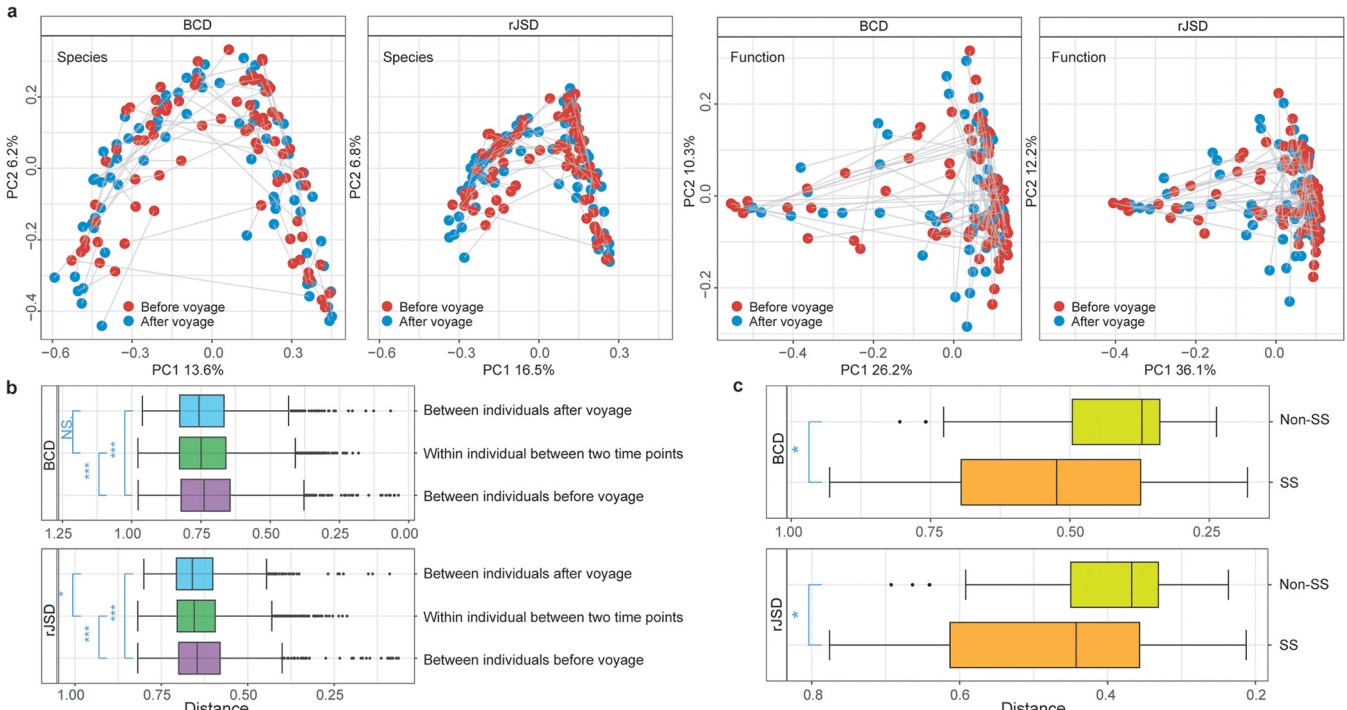

**FIG 2** Impact of voyage on the diversity of the gut microbiome. (a) Principal Coordinate Analysis based on both the taxonomical (distribution of microbial species generated by mOTUs2) and functional (metagenomic functions generated by HUMAnN2) profiles between two time points. The developing trajectory for each individual during the voyage was connected by gray lines. (b) The BCD and rJSD between individuals at the beginning and end of the voyage, and the distance within individual between two time points were compared. (c) Boxplots of the individual perturbations in the gut microbiome between SS and non-SS groups based on BCD and rJSD (BCD: $P = 0.024$, rJSD: $P = 0.021$).

dissimilarity [BCD]: $F = 1.762$, $P = 0.043$; rJSD: $F = 1.481$, $P = 0.063$). Notably, based on Permanova, no effect sizes for any symptoms were significant (Table 1), suggesting only a weak influence of any single symptom on the gut microbiome during the voyage. These results indicate that the crew members' gut microbiome is associated with a group of symptoms (i.e., SS instead of any single symptom) (Table 1).

Secondly, to investigate the perturbations during the voyage for each individual, we compared perturbations (change in microbial dynamics) in the gut microbiome between crew members suffering SS (individuals clustered in Group A, Fig. 1d) with non-SS (crew members that were not clustered in Group A, Fig. 1d). Although there was no significant difference in alpha diversity (Shannon index, Wilcoxon rank-sum Test, $P = 0.73$) and PCoA clustering (Fig. 2a) in the gut microbiome between day 1 and day 135, individual perturbation between the two time points (indicated by BCD and rJSD) showed a significant change in the gut microbiome (Fig. 2b). The difference between two time points in SS samples was significantly higher than that of non-SS samples (Wilcoxon rank-sum Test, BCD: $P = 0.024$; rJSD: $P = 0.021$, Fig. 2c). However, such correlation was not found between the gut microbiome and any single symptom. For example, crew members were grouped by symptoms (e.g., poor sleep quality versus symptomless), and the individual perturbations were grouped accordingly and compared, but no significant difference was observed for any single symptom (Fig. S1, Method, in the supplemental material).

**Microbial signature for SS during the voyage.** To identify the microbial biomarkers contributing to the development of SS, we compared the abundance of microbial species and functions between SS group (defined by Group A symptoms) and non-SS group from beginning to the end of the voyage (Fig. 3a; Method). Overall, 15 species were significantly increased (Wilcoxon rank-sum test, $P < 0.05$) and four species decreased in SS group compared to non-SS group during the voyage. The abundance of 25 gene families from 12 pathways were found to be decreased compared to

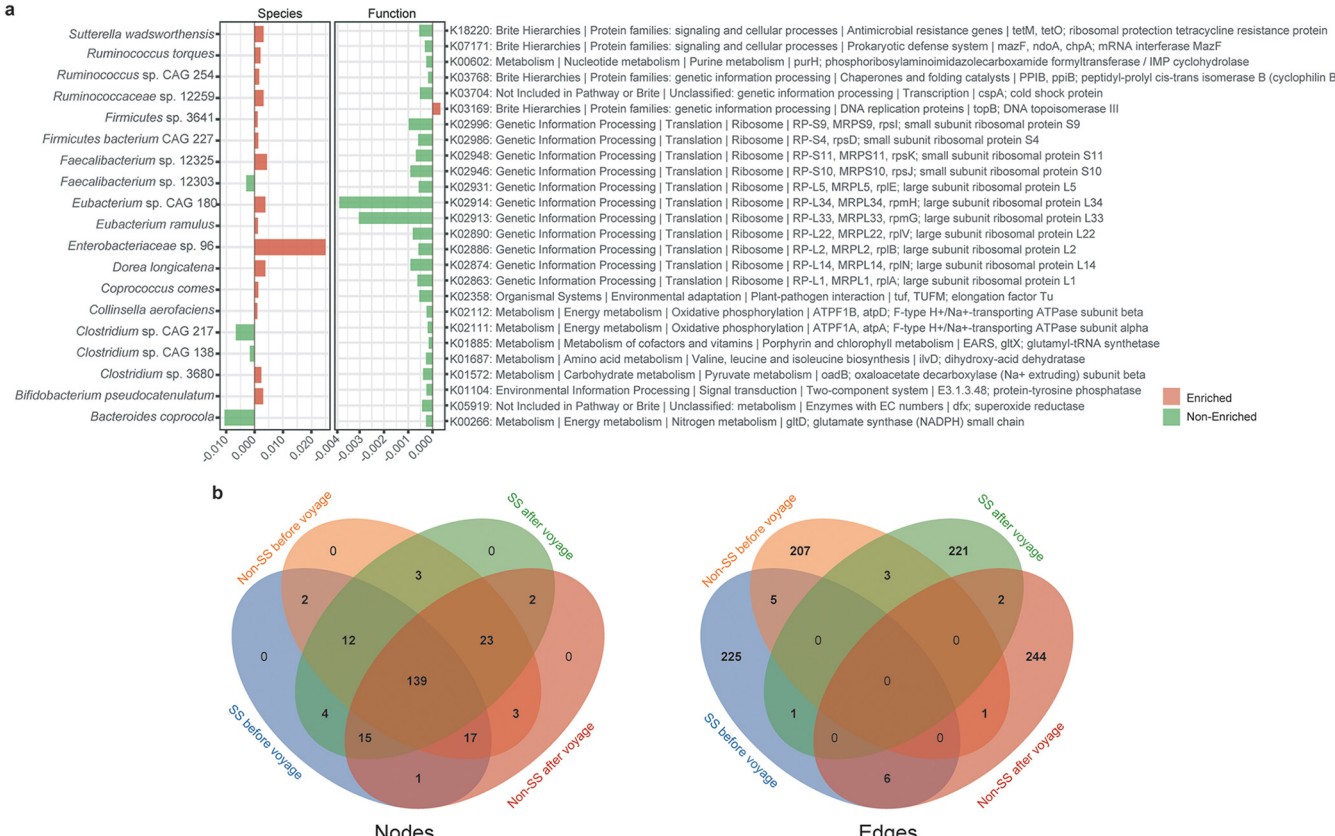

**FIG 3** Microbial signature of seafaring syndrome. (a) Bar plots of microbial species and functions that changed in abundance during the voyage (green represents species/functions that increased while red represents species/functions that decreased after the voyage). (b) Venn diagram for nodes (left panel) and edges (right panel) of four co-occurrence networks: SS and non-SS at beginning (day 1) and end (day 135) of the voyage.

non-SS group during the voyage, while one gene family was increased in SS group. This suggests that a microbial signature can be captured in SS crew members during the voyage, which can be quantified in 19 taxonomical and 26 functional biomarkers. Then, 19 differential species were searched against previous chronic disease microbiome studies, and the change of 17 species are found to be coordinated with the chronic diseases' outcome or development, suggesting the relevance between SS and chronic diseases (Table 2).

To better understand the microbial dynamics difference between SS and non-SS groups, co-occurrence network analysis was employed for the stool samples at before and after the voyage (Fig. S2, Table S1, Method, in the supplemental material). The following thresholds were used to compare nodes, edges, and densities of the four co-occurrence networks (Fig. 3b): relative abundance greater than 0.001, prevalence greater than 10%, and $P = 0.01$. We found SS and non-SS have unique microbial dynamics based on the co-occurrence network pattern (nodes and edges) and properties (density), and the correlation of microbial species in the two groups are different even at the beginning of the voyage.

**Prediction of SS before the voyage.** To explore whether SS can be predicted, Random Forest (RF) model was used to build a prediction model which distinguished SS group from non-SS groups based on the microbial taxonomic and functional profiling of samples taken before the voyage. Performance improvement was minimal once the top 28 most discriminatory species (based on the importance evaluated by RF) model and functions were included (Table S2 in the supplemental material; Fig. 4a). Ultimately, samples from SS could be distinguished from non-SS samples with 84.4% accuracy (10-fold cross validation area under the concentration-time curve [AUC] = 0.91, Fig. 4b). On the other hand, although RF models using baseline data from microbial species or microbial

**TABLE 2** Association of differential abundant taxa between symptomatic and asymptomatic crew members and chronic diseases

| Species | Associated with | Change | Reference |
|---|---|---|---|
| *Bacteroides coprocola* | Neurological disorders | Decrease | (38) |
| *Bifidobacterium pseudocatenulatum* | Type 2 diabetes | Increase | (39) |
| *Clostridium* sp. 3680 | Systemic lupus erythematosus | Increase | (40) |
| *Clostridium* sp. CAG 138 | Peanut allergy | Decrease | (41) |
| *Clostridium* sp. CAG 217 | | | |
| *Collinsella aerofaciens* | Irritable bowel syndrome | Increase | (42) |
| *Coprococcus comes* | Chronic widespread musculoskeletal pain | Decrease | (43) |
| *Dorea longicatena* | Overweight/obese | Increase | (44) |
| | Circadian rhythm disturbance | Increase | (45) |
| *Enterobacteriaceae* sp. 96 | *Clostridioides difficile* infection | Increase | (46) |
| *Eubacterium ramulus* | Metabolize quercetin of polyphenol-rich foods (fruit and vegetable) | NA | (47, 48) |
| *Eubacterium* sp. CAG 180 | Working memory of obese subjects | Increase | (49) |
| *Faecalibacterium* sp. 12303 | Food-allergy | Increase | (50) |
| *Faecalibacterium* sp. 12325 | | | |
| *Firmicutes bacterium* CAG 227 | Unhealthy diet | Increase | (51) |
| *Firmicutes* sp. 3641 | | | |
| *Ruminococcaceae* sp. 12259 | Mood | Increase | (52) |
| *Ruminococcus* sp. CAG 254 | Crohn disease | Increase | (53) |
| *Ruminococcus torques* | Obesity | Increase | (54) |
| | Irritable bowel syndrome | Increase | (42) |
| | Circadian rhythm disturbance | Increase | (55) |
| *Sutterella wadsworthensis* | Ulcerative colitis | Increase | (28) |

functions have comparable performance (accuracy = 80.5%, AUC = 0.84; accuracy = 83.1%, AUC = 0.84 separately), the RF model based on questionnaire answers could only predict the onset of SS with 68.8% (AUC = 0.69) accuracy (Fig. 4b). Thus, biomarkers from baseline gut microbiome data alone can potentially predict the likelihood of SS developing, without the need of determining physical or psychological status.

To validate SS prediction model, we employed another relevant cohort by Zhang et al., (23), where a 30-day longitudinal study was designed to explore the impacts of the sea voyage on the gut microbiome of sailors and whether probiotics can be a feasible approach for protecting gut health during the sea voyage. Based on the supply of a mixed probiotic product (containing *Lactobacillus* and *Bifidobacterium*), sailors were divided into control group (probiotic dose was not taken) and probiotic group (probiotic dose was taken). SS was predicted for 56 sailors using the microbiome data before voyage, and the prediction results were compared with outcome of SS (Fig. 4c and d). We

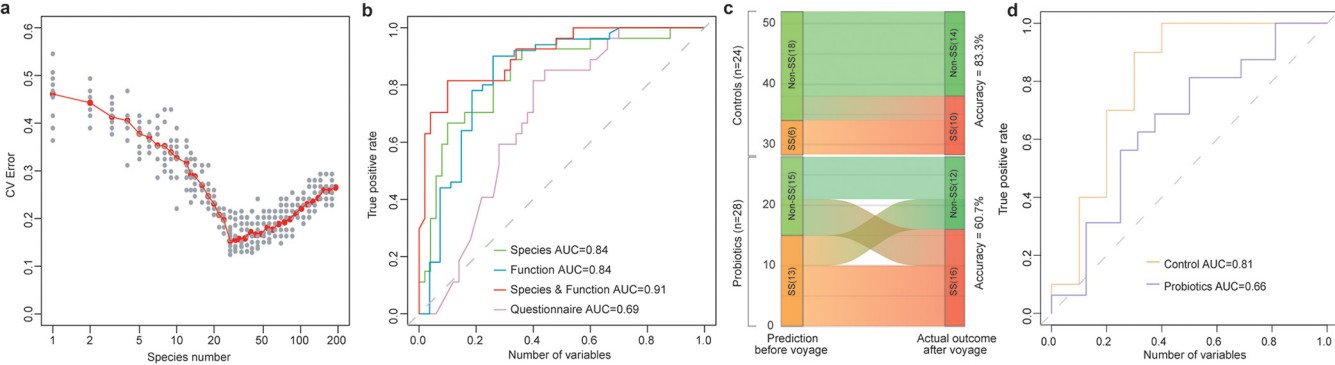

**FIG 4** Random Forest model for predicting the likelihood that crew members would develop seafaring syndrome. (a) Selection of biomarkers based on the gut microbiome for RF model to predict the likelihood of seafaring syndrome developing. The *x* axis refer to the feature (species) number used in the RF model, and the *y* axis stands for the error rate of the cross-validation. The relationship between the number of variables in the RF model and model performance were analyzed; 28 biomarkers with the most discriminating power were selected. (b) Prediction performance of RF models using different biomarkers at baseline (e.g.,, only microbial species, only microbial functions, only questionnaire before the voyage, and microbial species plus gene functions), as assessed via the Area Under the Receiver Operating Characteristic Curve (AUC). (c) Comparison of SS prediction result using the microbiome data from day 1 with the actual outcome of SS by questionnaires at the end of the voyage. (d) Performance of SS prediction model for the control and probiotic group of the 30-day voyage.

found that six sailors in the control group were predicted with SS and 18 individuals were predicted healthy, with 83.3% (AUC = 0.81) accuracy of the prediction.

## DISCUSSION

Questionnaires clearly indicate that the crew members showed a sub-optimal health status after the ocean voyage. However, none of the symptoms taken individually can positively relate to the gut microbiome since individuals can respond differently to the same stimulus and/or environment (24). This led us to consider that similar changes in the gut microbiome could trigger different physical and psychological responses. To test this hypothesis and to rule out the noise introduced by individual differences, we performed cluster analysis on the symptoms in the first place, which is rarely performed in previous studies due to sample size limitation and questionnaire design. For the first time to our knowledge, we identified the co-appearance of symptoms of abnormal defecation frequency, insomnia, poor sleep quality, nausea and overeating (symptoms in Group A) best described the sub-optimal health status of the majority of crew members (i.e., those identified as having 'Seafaring Syndrome (SS)'), and a strong association between SS and the gut microbiome was proved. Next, a microbial signature of SS was identified, which supports the hypothesis that similar changes in the gut microbiome can trigger both similar (e.g., symptoms in Group A) and different physical/psychological responses (e.g., symptoms in Group B or symptoms in Group C).

Moreover, to apply the concept of SS to real-time prediction of crew member response to extreme environments, we built a model that could predict the likelihood of SS. The model predicted SS with 84.4% accuracy (AUC = 0.91) before the voyage, which would be a useful health risk assessment for crew members and potential criteria for the selection of individuals during a long-term sea voyage. In addition, by employing a 30-day voyage cohort as validation, we further proved the effectiveness of the model with the addition of comparing the effect of probiotics on sailor health status. We found a decreasing accuracy of the model in the sailors intervened by probiotics, suggesting probiotics have the potential to prevent or mitigate SS. Studying the association between the gut microbiome and sub-optimal health status in extreme environments will give us a better understanding of the interactions between the gut microbiome health and dysbiosis that may be related to systemic or chronic diseases which is an important health problem in today's society (chronic diseases contribute to around 41 million (71%) of all the deaths globally) (25). This study provides a fundamental basis for discovering potential microbial and bacterial gene function predictors for the sub-optimal health status caused by extreme environments and demonstrates how pretreatment of symptoms could mitigate multi-symptom dysbiosis specific to each individual. We plan to introduce probiotics as intervention e.g., some well-studied probiotics such as *Lactobacillus casei* Zhang (26–28), *Lactobacillus plantarum* P-8 (29), *Bifidobacterium lactis* V9 (26), *Lactobacillus rhamnosus* Probio-M9 (29), and *Bifidobacterium lactis* Probio-M8 (30)] in the ocean voyage and dig out the interaction between the sub-optimal health status and the gut microbiome in our next work.

## MATERIALS AND METHODS

**Experimental design and subject recruitment.** This study included 77 crew members who participated in a 5-month long ocean voyage (135 days). Fecal samples were collected at the beginning (baseline at day 1; morning of the first day after boarding) and at the end (day 135, morning of the last day before landing) of the voyage. At the same time points, 24 physical and psychological states were defined for each participant using their responses to a questionnaire. Dietary records showed that the menu was repeated every week during the voyage; at least two kinds of nonstaple food and two or three kinds of fruits were guaranteed every day to meet the crew members' nutritional needs. Sample protector (CW0592M, CWBIO, China) was added to each stool sample in a ratio of 1:5 prior to storage at −20℃ until further processing.

**Questionnaire design.** The questionnaire consisted of three parts: (i) physical indicators, such as BMI, and incidence of: backache, headache, stomachache, pectoralgia, muscular soreness, dyspnea and palpitation; (ii) psychological indicators, including: insomnia, poor sleep quality, overeating, nausea, inactivity, poor appetite, feelings of loneliness, depression, self-accusation, and mistrustfulness; (iii) defecation-related indicators, including stool type and incidence of: bloody stools, incomplete defecation,

bowel difficulties, constipation, and defecation frequency (weekly). For more details about the definition of the above indicators, please refer to Table S3 in the supplemental material. Subject's responses were quantified by employing a scoring system based on symptom severity: 1, no symptom/low frequency/healthy; up to 3, moderate; up to 5, symptoms were frequent and severe. Scores for individual items were then used to rate symptom severity. Then, we categorized subjects into healthy (score 1–2) and symptomatic (score 3–5) based on the scores. For more details of the results of the questionnaire (Table S4 in the supplemental material).

**Clustering of symptoms.** The least variance method (Ward.d) from the R package 'pheatmap' (https://cran.r-project.org/web/packages/pheatmap/index.html) was used to cluster physical and psychological symptoms (except BMI) into groups by similarity in Ward's minimum distance method.

**Whole metagenomic sequencing.** Stool samples were thawed on ice for 1 h and three 1.5 ml bacterial suspensions from each (a total of 4.5 ml) were vortexed and used for DNA extraction. A standard Qiagen DNA Stool minikit (Qiagen, Hilden, Germany) was used to extract DNA following the manufacturer's instructions. Genomic DNA quality and concentration were analyzed by gel electrophoresis and Nanodrop8000 (Thermo Electron Corp., Waltham, MA, USA), r espectively. The final DNA concentration was above 100 ng/$\mu$l and the 260 nm/280 nm ratio was between 1.8 and 2.0. All samples were sequenced using Illumina HiSeq 2500. After quality control and human DNA removal a mean of 22,255,766 high-quality end reads were obtained for each sample.

**Taxonomical and functional profiling of WMS data and statistical analysis.** We used mOTUs2 (31) and HUMAnN2 (32) on high-quality end reads for taxonomical and functional profiling with default parameters (33). In the differential abundance analysis, the Wilcoxon signed-rank test was used to identify significant changes in microbial species and gene functions between the two time points (at the beginning and end of the voyage) for both 'symptomatic' and 'symptomless' crew members employing (cut-off $P$ value = 0.05). Species and gene functions that coordinately increased/decreased in both groups were considered influenced by time points rather than the voyage itself and then were discarded. The remaining species and functions were considered as microbial signatures of symptoms.

**Association between symptoms and the gut microbiome.** Firstly, BCD and rJSD were used to measure changes in the gut microbiome of an individual between the beginning (baseline) and the end of the voyage; and termed this as "individual perturbation." Secondly, for each symptom, the crew were divided into symptomless and symptom groups as described before. Thirdly, BCD or rJSD values for each symptom were compared between the symptomless and symptom groups using the Wilcoxon test. All statistical analysis was done using the R script in Parallel-META-3.5 (34).

**Microbial signature and co-occurrence network analysis.** For the crew members in SS group, the Wilcoxon signed-rank test was used to identify microbial species and functions that changed in abundance, respectively, between the beginning and the end of the voyage. We then repeated this using the same method for non-SS group. The species and functions that had changed between the beginning and end of the voyage were then compared between SS and non-SS groups; species and functions that changed in the same way in both groups were considered as an effect of time point rather than the voyage and were discarded from analysis. The remaining species and functions in SS group were considered as the microbial signature of SS development. Then, MetagenoNets (35) was employed for co-occurrence network analysis based on SparCC correlation inference algorithms. The co-occurrence network was trimmed by taking the relative abundance > 0.001, prevalence > 10%, and $P$ = 0.01 as the threshold.

**RF model construction.** The top-ranking AD-discriminatory taxa that led to reasonably good fit were identified based on "rfcv" function in the 'randomForest' package (36, 37). RF models were then trained to classify SS in the training set which included samples from SS and non-SS groups using both taxonomy and function profiles. The results were evaluated with a 10-fold cross-validation approach, and model performance was evaluated by receiver operating characteristics. Default parameters of the RF were applied (ntree = 5,000, using default mtry of p/3, where p is the number of input taxa).

**Ethics approval and consent to participate.** All procedures performed in studies involving human participants were in accordance with the ethical standards of the institutional and/or national research committee and with the 1964 Helsinki declaration and its later amendments or comparable ethical standards.

**Data availability.** Data are available in a public, open access repository. All sequence data from this study has been submitted to Sequence Read Archive (https://www.ncbi.nlm.nih.gov/sra) and can be accessed through the BioProject IDs: PRJNA788972, SRP350954.

## SUPPLEMENTAL MATERIAL

Supplemental material is available online only.
**SUPPLEMENTAL FILE 1**, PDF file, 0.6 MB.
**SUPPLEMENTAL FILE 2**, XLSX file, 0.02 MB.

## ACKNOWLEDGMENTS

We sincerely thank all the volunteers for their participation.

This work was supported by the National Natural Science Foundation of China (31720103911); China Agriculture Research System of MOF and MARA; and Inner Mongolia Science and Technology Major Projects (2021ZD0014).

H.Z., Z.H.S., and Z.S., designed the study. J.Z., Y.J., H.C., and M.L., collected the samples and conducted the experiments. Z.S., M.Z., and M.L., analyzed the data and prepared the manuscript. Y.B. provided suggestions and modified the manuscript. All authors approved the final version of the manuscript.

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
