## [Reviewer comments · Microbiology Spectrum]

Microbiology Spectrum

Interactions between human gut microbiome dynamics and sub-optimal health symptoms during seafaring expeditions

Zheng Sun, Meng Zhang, Min Li, Yogendra Bhaskar, Jinshan Zhao, Youran Ji, Hongbing Cui, Heping Zhang, and zhihong sun

Corresponding Author(s): zhihong sun, Inner Mongolia Agricultural University

Review Timeline:

Submission Date:	July 15, 2021
Editorial Decision:	September 7, 2021
Revision Received:	November 3, 2021
Editorial Decision:	December 8, 2021
Revision Received:	December 17, 2021
Accepted:	December 20, 2021

Editor: Adelumola Oladeinde

Reviewer(s): Disclosure of reviewer identity is with reference to reviewer comments included in decision letter(s). The following individuals involved in review of your submission have agreed to reveal their identity: Maite Ghazaleh (Reviewer #2)

Transaction Report:

DOI: <https://doi.org/10.1128/Spectrum.00925-21>

September 7, 2021

Prof. zhihong sun
Inner Mongolia Agricultural University
Key Laboratory of Dairy Biotechnology and Engineering, Ministry of Education, School of Food Science and Engineering
306 Zhaowuda Road
Huhhot, Inner Mongolia 010018
China

Re: Spectrum00925-21 (Microbial Signature of Ocean-Going Syndrome)

Dear Prof. zhihong sun:

Thank you for submitting your manuscript to Microbiology Spectrum. Reviewer 1 has raised critical comments that require a robust modification of the results presented. In your rebuttal, please address these comments sufficiently (either by adding new data or text) before it can be sent for a re-review. When submitting the revised version of your paper, please provide (1) point-by-point responses to the issues raised by the reviewers as file type "Response to Reviewers," not in your cover letter, and (2) a PDF file that indicates the changes from the original submission (by highlighting or underlining the changes) as file type "Marked Up Manuscript - For Review Only". Please use this link to submit your revised manuscript - we strongly recommend that you submit your paper within the next 60 days or reach out to me. Detailed information on submitting your revised paper are below.

Link Not Available

Sincerely,

Adelumola Oladeinde

Journals Department
Reviewer comments:

Reviewer #1 (Comments for the Author):

The Manuscript entitled "Microbial Signature of Ocean-Going Syndrome" describes microbiota associations with symptoms of a stress/health disorder called "Ocean-Going Syndrome". The manuscript was well written and easy to read. I am happy to see an attempt to connect complex phenotypic data with microbial compositional data. However, there are a number of methodological issues that are problematic. It's unclear how much of the microbial changes observed over the period at sea is due to the unique conditions (stress) of being at sea, or simply due to the fact that the individuals sampled leave their home environments (different social settings, cooking different food) and housed in close proximity (similar microbial environments) and fed them the same food. Observed changes may not be stress induced. Samples should have been collected when symptoms first arose, not just at the end.

The questionnaire should be provided as a supplemental item. It seems that self-assessment could be subjective and highly variable. How were these data quantitated?

My biggest concern is how the data were analyzed. No information was provided to explain how false discovery rates were

addressed. The NCBI data link doesn't work, so I couldn't check the data myself.

By only including the data that were different within the treatment groups for analysis seems biased. What were the result of comparisons between sick and not sick individuals? The manuscript doesn't explore the specific taxonomic and functional differences, which is a missed opportunity to improve our understanding of changes at that resolution. More transparency and description is needed for understanding and confidence of the findings highlighted in this manuscript.

Reviewer #2 (Comments for the Author):

Please see the attached document. Great study, but needs work (mainly in writing) to clarify conclusions.

Staff Comments:

Preparing Revision Guidelines

Please return the manuscript within 60 days; if you cannot complete the modification within this time period, please contact me. If you do not wish to modify the manuscript and prefer to submit it to another journal, please notify me of your decision immediately so that the manuscript may be formally withdrawn from consideration by Microbiology Spectrum.

Adelumola Oladeinde, PhD, Editor, Microbiology Spectrum

November 3rd, 2021

Dear Dr. Oladeinde,

Thank you and the reviewers for commenting on our manuscript and giving us the opportunity to revise. Here we have modified the manuscript and addressed all the comments as detailed below.

Editorial Comments

Thank you for submitting your manuscript to Microbiology Spectrum. Reviewer 1 has raised critical comments that require a robust modification of the results presented. In your rebuttal, please address these comments sufficiently (either by adding new data or text) before it can be sent for a re-review. When submitting the revised version of your paper, please provide (1) point-by-point responses to the issues raised by the reviewers as file type "Response to Reviewers," not in your cover letter, and (2) a PDF file that indicates the changes from the original submission (by highlighting or underlining the changes) as file type "Marked Up Manuscript - For Review Only". Please use this link to submit your revised manuscript - we strongly recommend that you submit your paper within the next 60 days or reach out to me. Detailed information on submitting your revised paper are below.

Response 1: We greatly appreciate the opportunity to further improve our manuscript. In the revised manuscript, we have verified the statistical methods as well as the accuracy of the values in the tables and figures. All revisions in the manuscript are highlighted, and the file is submitted as "Marked Up Manuscript - For Review Only.doc". Our point-by-point responses are itemized below.

Responses to Reviewer #1

Comments for the Author

The Manuscript entitled "Microbial Signature of Ocean-Going Syndrome" describes microbiota associations with symptoms of a stress/health disorder called "Ocean-Going Syndrome". The manuscript was well written and easy to read. I am happy to see an attempt to connect complex phenotypic data with microbial compositional data. However, there are a number of methodological issues that are problematic. It's unclear how much of the microbial changes observed over the period at sea is due to the unique conditions (stress) of being at sea, or simply due to the fact that the individuals sampled leave their home environments (different social settings, cooking different food) and housed in close proximity (similar microbial environments) and fed them the same food. Observed changes may not be stress induced. Samples should have been collected when symptoms first arose, not just at the end.

Response 2: We thank Reviewer #1 very much for reviewing our manuscript and the positive comments and interest in our work. As the Reviewer #1 mentioned, we agree that there are multiple factors that could lead to changes in the gut microbiome, e.g., stress, food, and environment. However, instead of studying the association between the gut microbiome and stress, we aimed to understand its correlation with the different chronic disease-like-symptoms.

Previous studies have demonstrated the close association between the gut microbiome and many chronic diseases [1-10]; similar symptoms of these chronic diseases happen to be commonly seen in crew members on long voyage [11-16]. This raises the question of whether the long voyage could be considered as an accelerator for chronic disease onset. Indeed, this was our motivation of tracking changes in gut microbiome in long voyage. In addition, since stress, food, and environment all have been proven to be significant factors driving changes in the gut microbiome [17-22], we sought to consider the gut microbiome as a black box that bridges environmental factors with chronic diseases (**Figure R1**).

Figure R1. The motivation for studying changes in human gut microbiome during long voyage. Various factors can impact the gut microbiome, and we propose that the gut microbiome serves as messengers reflecting environmental stressors in the form of chronic disease symptoms under a sub-optimal health state. Therefore, we aim to test if long voyage could accelerate the onset and progression of chronic diseases.

In summary, studying the association between the gut microbiome and sub-optimal health state in extreme environments would give us a better understanding of interactions between the gut microbiome and dysbacteriosis related diseases (systemic or chronic), which is an important health concern in today's society.

The questionnaire should be provided as a supplemental item. It seems that self-assessment could be subjective and highly variable. How were these data quantitated?

Response 3: We thank Reviewer #1 for this excellent comment. In our original manuscript, we provided the questions in the questionnaires in supplementary Table S3. As suggested, we added a new supplementary table (Table S4) showing all subjects' responses to the questionnaire for before and after the voyage.

“Subject’s responses were quantified by employing a scoring system based on symptom severity: 1, no symptom/low frequency/healthy; up to 3, moderate; up to 5, symptoms were frequent and severe. Scores for individual items were then used to rate symptom severity. Then, we categorized subjects into healthy (score 1-2) and symptomatic (score 3-5) based on the scores”. The description above has been added in the method section under “Questionnaire design”.

My biggest concern is how the data were analyzed. No information was provided to explain how false discovery rates were addressed. The NCBI data link doesn't work, so I couldn't check the data myself.

Response 4: Thank you for your comment. We apologize that the methods for data analysis were not adequately described. Now we have added a new paragraph in the method section entitled “Taxonomical and functional profiling of WMS data and statistical analysis” to explain the data processing and analysis. Please also refer to our Response #5.

As for the NCBI data link, we are sorry that it did not work when reviewing this manuscript. We assume this is because we set a release date in the end of this year (Dec 2021). Now we have released the data directly, and the raw sequencing data is now accessible (PRJNA629464, SRP259900, access link: https://www.ncbi.nlm.nih.gov/Traces/study/?acc=PRJNA629464&o=acc_s%3Aa).

By only including the data that were different within the treatment groups for analysis seems biased. What were the result of comparisons between sick and not sick individuals? The manuscript doesn't explore the specific taxonomic and functional differences, which is a missed opportunity to improve our understanding of changes at that resolution. More transparency and description is needed for understanding and confidence of the findings highlighted in this manuscript.

Response 5: We thank Reviewer #1 for this critical comment. We apologize that the writing of the manuscript confused Reviewer #1. We did conduct the comparison between sick and not sick individuals in our manuscript.

Since we have time series data at two time points, we adjusted the general comparison method as: “In the differential abundance analysis, the Wilcoxon signed-rank test was used to identify significant changes in microbial species and gene functions between the two time points (at the beginning and end of the voyage) for both ‘symptomatic’ and ‘symptomless’ crew members employing (cut-off p value = 0.05). Species and gene functions that coordinately increased/decreased in both groups were considered influenced by time points rather than the voyage itself and then were discarded. The remaining species and functions were considered as microbial signatures of symptoms.” The description of this statistical analysis has been added in the Method section of the revised manuscript.

We performed this analysis for the syndrome we discovered as we have stated in the revised manuscript: “To identify the microbial biomarkers contributing to the development of the seafaring syndrome (SS), we compared the abundance of microbial species and functions between SS group (defined by Group A symptoms) and non-SS group from beginning to the end of the voyage (Fig. 3a; Method). Overall, 15 species were significantly increased (Wilcoxon rank-sum test, $p < 0.05$) and four species decreased in SS group compared to non-SS group during the voyage. The abundance of 25 gene families from 12 pathways were found to be decreased compared to non-SS group during the voyage, while one gene family was increased in SS group.”

To consolidate our findings, our revised manuscript provides literature search results of these 19 differential taxa that were previously reported to be associated or responsive to chronic diseases in gut microbiome studies (**Table R1**, Table 2 in the revised manuscript). The manuscript was revised accordingly. We found that 17 of the 19 species showed coordinated changes in previous chronic disease related studies, suggesting the relevance between the syndrome we discovered with chronic disease.

Table R1 (Table 2 in the revised manuscript) Association of differential abundant taxa between symptomatic and asymptomatic crew members and chronic diseases.

Species	Associated with	Change	Reference
Bacteroides coprocola	Neurological disorders	Decrease	[23]
Bifidobacterium pseudocatenulatum	Type 2 diabetes	Increase	[24]
Clostridium sp. 3680	Systemic lupus erythematosus	Increase	[25]
Clostridium sp. CAG 138	Peanut allergy	Decrease	[26]
Clostridium sp. CAG 217			
Collinsella aerofaciens	Irritable bowel syndrome	Increase	[27]
Coprococcus comes	Chronic widespread musculoskeletal pain	Decrease	[28]
Dorea longicatena	Overweight/Obese	Increase	[29]

	Circadian rhythm disturbance	Incr ease	[30]
Enterobacteriaceae sp. 96	Clostridioides difficile infection	Incr ease	[31]
Eubacterium ramulus	Metabolize quercetin of polyphenol-rich foods (fruit and vegetable)	NA	[32, 33]
Eubacterium sp. CAG 180	Working memory of obese subjects	Incr ease	[34]
Faecalibacterium sp. 12303	Food allergy	Incr	[35]
Faecalibacterium sp. 12325		ease	
Firmicutes bacterium CAG 227	Unhealthy diet	Incr	[36]
Firmicutes sp. 3641		ease	
Ruminococcaceae sp. 12259	Mood	Incr ease	[37]
Ruminococcus sp. CAG 254	Crohn disease	Incr ease	[38]
Ruminococcus torques	Obesity	Incr ease	[39]
	Irritable bowel syndrome	Incr ease	[27]
	Circadian rhythm disturbance	Incr ease	[40]
Sutterella wadsworthensis	Ulcerative colitis	Incr ease	[41]

Responses to Reviewer #2

Please see the attached document. Great study, but needs work (mainly in writing) to clarify conclusions.

Response 6: We thank Reviewer #2 very much for reviewing our manuscript. We have addressed each of the comments in a separate file named “Response to Reviewer #2.doc”. To keep all the comments from Reviewer #2 and our responses in the document, changes were made using the “track change” function in “Response to Reviewer #2.doc”. The “Marked Up Manuscript For Review Only.doc” showed the final version.

Reference cited in this response letter:

1. Du Toit, A., *The gut microbiome and mental health*. Nat Rev Microbiol, 2019. **17**(4): p. 196.
2. Thaiss, C.A., *Microbiome dynamics in obesity*. Science, 2018. **362**(6417): p. 903-904.
3. de Steenhuijsen Piters, W.A.A., J. Binkowska, and D. Bogaert, *Early Life Microbiota and Respiratory Tract Infections*. Cell Host Microbe, 2020. **28**(2): p. 223-232.
4. Jie, Z., et al., *The gut microbiome in atherosclerotic cardiovascular disease*. Nat Commun, 2017. **8**(1): p. 845.
5. Black, C.J., et al., *Functional gastrointestinal disorders: advances in understanding and management*. Lancet, 2020. **396**(10263): p. 1664-1674.
6. Ticinesi, A., et al., *Understanding the gut-kidney axis in nephrolithiasis: an analysis of the gut microbiota composition and functionality of stone formers*. Gut, 2018. **67**(12): p. 2097-2106.
7. Wei, Y., et al., *Alterations of gut microbiome in autoimmune hepatitis*. Gut, 2020. **69**(3): p. 569-577.
8. Harrison, C.A. and D. Taren, *How poverty affects diet to shape the microbiota and chronic disease*. Nature Reviews Immunology, 2018. **18**(4): p. 279-287.
9. Zachariassen, L.F., et al., *Sensitivity to oxazolone induced dermatitis is transferable with gut microbiota in mice*. Scientific

- Reports, 2017. **7**.
10. Guo, Z., et al., *Intestinal Microbiota Distinguish Gout Patients from Healthy Humans*. Scientific Reports, 2016. **6**.
 11. Srivastava, A.K., et al., *Probiotics maintain the gut microbiome homeostasis during Indian Antarctic expedition by ship*. Scientific Reports, 2021. **11**(1).
 12. Merwin, M.R. and F.M. Ochberg, *The long voyage: policies for progress in mental health*. Health Aff (Millwood), 1983. **2**(4): p. 96-127.
 13. Iversen, R.T., *The mental health of seafarers*. Int Marit Health, 2012. **63**(2): p. 78-89.
 14. Xie, S., et al., *Analysis and determinants of Chinese navy personnel health status: a cross-sectional study*. Health Qual Life Outcomes, 2018. **16**(1): p. 138.
 15. Zhang, J., et al., *Probiotics maintain the intestinal microbiome homeostasis of the sailors during a long sea voyage*. Gut Microbes, 2020. **11**(4): p. 930-943.
 16. Visser, J.T., *Patterns of illness and injury on Antarctic research cruises, 2004-2019: a descriptive analysis*. J Travel Med, 2020. **27**(6).
 17. Rothschild, D., et al., *Environment dominates over host genetics in shaping human gut microbiota*. Nature, 2018. **555**(7695): p. 210-215.
 18. Zhang, C., et al., *Structural resilience of the gut microbiota in adult mice under high-fat dietary perturbations*. ISME J, 2012. **6**(10): p. 1848-57.
 19. Turroni, S., et al., *Temporal dynamics of the gut microbiota in people sharing a confined environment, a 520-day ground-based space simulation, MARS500*. Microbiome, 2017. **5**(1): p. 39.
 20. Brooks, J.F., 2nd, et al., *The microbiota coordinates diurnal rhythms in innate immunity with the circadian clock*. Cell, 2021. **184**(16): p. 4154-4167 e12.
 21. Wu, W.L., et al., *Microbiota regulate social behaviour via stress response neurons in the brain*. Nature, 2021. **595**(7867): p. 409-414.
 22. Xu, C., et al., *The Gut Microbiome Regulates Psychological-Stress-Induced Inflammation*. Immunity, 2020. **53**(2): p. 417-428 e4.
 23. Grochowska, M., T. Laskus, and M. Radkowski, *Gut Microbiota in Neurological Disorders*. Arch Immunol Ther Exp (Warsz), 2019. **67**(6): p. 375-383.
 24. Zhao, L., et al., *Gut bacteria selectively promoted by dietary fibers alleviate type 2 diabetes*. Science, 2018. **359**(6380): p. 1151-1156.
 25. Chen, B.D., et al., *An Autoimmunogenic and Proinflammatory Profile Defined by the Gut Microbiota of Patients With Untreated Systemic Lupus Erythematosus*. Arthritis Rheumatol, 2021. **73**(2): p. 232-243.
 26. He, Z., et al., *Increased diversity of gut microbiota during active oral immunotherapy in peanut-allergic adults*. Allergy, 2021. **76**(3): p. 927-930.
 27. Malinen, E., et al., *Association of symptoms with gastrointestinal microbiota in irritable bowel syndrome*. World J Gastroenterol, 2010. **16**(36): p. 4532-40.
 28. Freidin, M.B., et al., *An association between chronic widespread pain and the gut microbiome*. Rheumatology (Oxford), 2021. **60**(8): p. 3727-3737.
 29. Companys, J., et al., *Gut Microbiota Profile and Its Association with Clinical Variables and Dietary Intake in Overweight/Obese and Lean Subjects: A Cross-Sectional Study*. Nutrients, 2021. **13**(6).
 30. Mortas, H., S. Bilici, and T. Karakan, *The circadian disruption of night work alters gut microbiota consistent with elevated risk for future metabolic and gastrointestinal pathology*. Chronobiol Int, 2020. **37**(7): p. 1067-1081.
 31. Haifer, C., et al., *Long-Term Bacterial and Fungal Dynamics following Oral Lyophilized Fecal Microbiota Transplantation in Clostridioides difficile Infection*. mSystems, 2021. **6**(1).
 32. Moco, S., F.P. Martin, and S. Rezzi, *Metabolomics view on gut microbiome modulation by polyphenol-rich foods*. J Proteome Res, 2012. **11**(10): p. 4781-90.
 33. Correa, T.A.F., et al., *The Two-Way Polyphenols-Microbiota Interactions and Their Effects on Obesity and Related Metabolic Diseases*. Frontiers in Nutrition, 2019. **6**.
 34. Arrioriaga-Rodriguez, M., et al., *Obesity Impairs Short-Term and Working Memory through Gut Microbial Metabolism of Aromatic Amino Acids*. Cell Metabolism, 2020. **32**(4): p. 548-+.
 35. Kourosh, A., et al., *Fecal microbiome signatures are different in food-allergic children compared to siblings and healthy children*. Pediatric Allergy and Immunology, 2018. **29**(5): p. 545-554.
 36. Asnicar, F., et al., *Microbiome connections with host metabolism and habitual diet from 1,098 deeply phenotyped individuals*. Nature Medicine, 2021. **27**(2): p. 321-+.
 37. Li, L., et al., *Gut microbes in correlation with mood: case study in a closed experimental human life support system*. Neurogastroenterology and Motility, 2016. **28**(8): p. 1233-1240.
 38. Ricanek, P., et al., *Gut bacterial profile in patients newly diagnosed with treatment-naive Crohn's disease*. Clin Exp Gastroenterol, 2012. **5**: p. 173-86.
 39. Wu, Q., et al., *Intestinal hypoxia-inducible factor 2alpha regulates lactate levels to shape the gut microbiome and alter thermogenesis*. Cell Metab, 2021. **33**(10): p. 1988-2003 e7.
 40. Deaver, J.A., S.Y. Eum, and M. Toborek, *Circadian Disruption Changes Gut Microbiome Taxa and Functional Gene Composition*. Frontiers in Microbiology, 2018. **9**.
 41. Gryaznova, M.V., et al., *Study of microbiome changes in patients with ulcerative colitis in the Central European part of Russia*. Heliyon, 2021. **7**(3): p. e06432.

Key Laboratory of Dairy Biotechnology and Engineering, Ministry of Education

Inner Mongolia Agricultural University

Mailing: 306 Zhaowuda Road, Huhhot, Inner Mongolia, China, 010018

Tel: +86-471-4305357; Fax: +86-471-4305357

Again, thanks to the anonymous reviewers for the constructive comments that have improved our manuscript. If you have any additional suggestions, please do not hesitate to contact me via phone call or email.

Zhihong Sun, Ph.D,

the National Science Fund for Excellent Young Scholars

Professor, Key Laboratory of Dairy Biotechnology and Engineering, Ministry of Education, Inner Mongolia Agricultural University, China

Inner Mongolia Agricultural University, 306 Zhaowuda Rd, Huhhot, Inner Mongolia, China

Email: sunzhihong78@163.com

Tel: (86) 0471-4308703, Fax: (86) 0471-4305357

December 8, 2021

Prof. zhihong sun
Inner Mongolia Agricultural University
Key Laboratory of Dairy Biotechnology and Engineering, Ministry of Education, School of Food Science and Engineering
306 Zhaowuda Road
Huhhot, Inner Mongolia 010018
China

Re: Spectrum00925-21R1 (Interactions between human gut microbiome dynamics and sub-optimal health symptoms during seafaring expeditions)

Dear Prof. zhihong sun:

Thank you for submitting your manuscript to Microbiology Spectrum. As you will see your paper is very close to acceptance. Please modify the manuscript along the lines recommended by the reviewers. As these revisions are quite minor, I expect that you should be able to turn in the revised paper in less than 30 days, if not sooner. If your manuscript was reviewed, you will find the reviewers' comments below.

When submitting the revised version of your paper, please provide (1) point-by-point responses to the issues I raised in your cover letter, and (2) a PDF file that indicates the changes from the original submission (by highlighting or underlining the changes) as file type "Marked Up Manuscript - For Review Only". Please use this link to submit your revised manuscript. Detailed instructions on submitting your revised paper are below.

Link Not Available

Sincerely,

Adelumola Oladeinde

Reviewer comments:

Reviewer #1 (Public repository details (Required)):

Data was found in GenBank but the metadata needs correcting. Samples within the ocean voyage bioproject were assigned to other host species: humans, *Mus musculus*, *Caenorhabditis elegans* and others. This needs to be checked and corrected.

Reviewer #1 (Comments for the Author):

Thank you for the attention to the reviewer comments and edits.

Reviewer #2 (Comments for the Author):

Supplemental material

Figure S2

Line 31

Provide figure legend details. When I look at the networks, what are the network components (dots, lines, clustering, etc.)?

Supplemental material
Table S4
Line 10
Is this table missing?

Preparing Revision Guidelines

- point-by-point responses to the issues I raised in your cover letter
- Upload a compare copy of the manuscript (without figures) as a "Marked-Up Manuscript" file.
- Each figure must be uploaded as a separate file, and any multipanel figures must be assembled into one file.
- Manuscript: A .DOC version of the revised manuscript
- Figures: Editable, high-resolution, individual figure files are required at revision, TIFF or EPS files are preferred

Please return the manuscript within 60 days; if you cannot complete the modification within this time period, please contact me. If you do not wish to modify the manuscript and prefer to submit it to another journal, please notify me of your decision immediately so that the manuscript may be formally withdrawn from consideration by Microbiology Spectrum.

Adelumola Oladeinde, PhD, Editor, *Microbiology Spectrum*

December 17, 2021

Dear Dr. Oladeinde,

We are glad to know that our work can be potentially acceptable for publication in *Journal* with adequate revision. Here we have modified the manuscript and addressed all the comments as detailed below.

Editorial Comments

*Thank you for submitting your manuscript to *Microbiology Spectrum*. As you will see your paper is very close to acceptance. Please modify the manuscript along the lines recommended by the reviewers. As these revisions are quite minor, I expect that you should be able to turn in the revised paper in less than 30 days, if not sooner. If your manuscript was reviewed, you will find the reviewers' comments below.*

When submitting the revised version of your paper, please provide (1) point-by-point responses to the issues I raised in your cover letter, and (2) a PDF file that indicates the changes from the original submission (by highlighting or underlining the changes) as file type "Marked Up Manuscript - For Review Only". Please use this link to submit your revised manuscript. Detailed instructions on submitting your revised paper are below.

*Thank you for the privilege of reviewing your work. Below you will find instructions from the *Microbiology Spectrum* editorial office and comments generated during the review.*

Response 1: As advised, all revisions in the manuscript are highlighted and submitted as “Marked Up Manuscript - For Review Only.doc”. Our point-by-point responses are itemized below.

Responses to Reviewer #1

Data was found in GenBank but the metadata needs correcting. Samples within the ocean voyage bioproject were assigned to other host species: humans, Mus musculus, Caenorhabditis elegans and others. This needs to be checked and corrected.

Response 2: We thank Reviewer #1 very much for pointing out the issue of metadata. To correct the metadata information, we recreated a new project number (PRJNA788972) for our manuscript and moved the raw sequencing data to SRP350954. Now the data has been released and can be accessed by: https://www.ncbi.nlm.nih.gov/Traces/study/?acc=PRJNA788972%20&o=acc_s%3Aa.

Responses to Reviewer #2

Supplemental material

Figure S2

Line 31

Provide figure legend details. When I look at the networks, what are the network components (dots, lines, clustering, etc.)?

Response 3: We thank Reviewer #2 for the constructive comment. As advised, we have supplemented some details in the legend of Figure S2 in the revised Supplementary Materials, now it reads:

“Figure S2. Co-occurrence networks of SS and non-SS at beginning (day 1) and end (day 135) of the voyage. Four co-occurrence networks of SS and non-SS at the beginning (day 1) and end (day 135) of the voyage are separately illustrated, which are based on SparCC correlation inference algorithms. The nodes in the network refer to the identified species with the size of the nodes indicating their relative abundance; The edges between nodes represent correlations between the nodes they connect.”

Supplemental material

Table S4

Line 10

Is this table missing?

Response 4: We thank Reviewer #2 for raising this concern. Table S4 was uploaded as a separate file in the system because it's too large to be added in the “Supplementary Materials.doc”.

Key Laboratory of Dairy Biotechnology and Engineering, Ministry of Education

Inner Mongolia Agricultural University

Mailing: 306 Zhaowuda Road, Huhhot, Inner Mongolia, China, 010018

Tel: +86-471-4305357; Fax: +86-471-4305357

Again, thanks to the anonymous reviewers for the constructive comments. If you have any additional suggestions, please do not hesitate to contact me via phone call or email.

Zhihong Sun, Ph.D,

the National Science Fund for Excellent Young Scholars

Professor, Key Laboratory of Dairy Biotechnology and Engineering, Ministry of Education, Inner Mongolia Agricultural University, China

Inner Mongolia Agricultural University, 306 Zhaowuda Rd, Huhhot, Inner Mongolia, China

Email: sunzhihong78@163.com

Tel: (86) 0471-4308703, Fax: (86) 0471-4305357

December 20, 2021

Prof. zhihong sun
Inner Mongolia Agricultural University
Key Laboratory of Dairy Biotechnology and Engineering, Ministry of Education, School of Food Science and Engineering
306 Zhaowuda Road
Huhhot, Inner Mongolia 010018
China

Re: Spectrum00925-21R2 (Interactions between human gut microbiome dynamics and sub-optimal health symptoms during seafaring expeditions)

Dear Prof. zhihong sun:

Your manuscript has been accepted, and I am forwarding it to the ASM Journals Department for publication. You will be notified when your proofs are ready to be viewed.

Sincerely,

Adelumola Oladeinde
Editor, Microbiology Spectrum

Journals Department
Supplemental file 1: Accept
Supplemental file2: Accept